# Consistency of Constrained Spectral Clustering under Graph Induced Fair Planted Partitions

**Shubham Gupta** *
IBM Research Paris-Saclay
Orsay 91400, France.
shubhamg@iisc.ac.in

**Ambedkar Dukkipati**
Computer Science and Automation
Indian Institute of Science, Bangalore, India.
ambedkar@iisc.ac.in

## Abstract

Spectral clustering is popular among practitioners and theoreticians alike. While performance guarantees for spectral clustering are well understood, recent studies have focused on enforcing "fairness" in clusters, requiring them to be "balanced" with respect to a categorical sensitive node attribute (e.g. the race distribution in clusters must match the race distribution in the population). In this paper, we consider a setting where sensitive attributes indirectly manifest in an auxiliary *representation graph* rather than being directly observed. This graph specifies node pairs that can represent each other with respect to sensitive attributes and is observed in addition to the usual *similarity graph*. Our goal is to find clusters in the similarity graph while respecting a new individual-level fairness constraint encoded by the representation graph. We develop variants of unnormalized and normalized spectral clustering for this task and analyze their performance under a *fair* planted partition model induced by the representation graph. This model uses both the cluster membership of the nodes and the structure of the representation graph to generate random similarity graphs. To the best of our knowledge, these are the first consistency results for constrained spectral clustering under an individual-level fairness constraint. Numerical results corroborate our theoretical findings.

## 1 Introduction

Consider a recommendation service that groups news articles by finding clusters in a graph which connects these articles via cross-references. Unfortunately, as cross-references between articles with different political viewpoints are uncommon, this service risks forming ideological filter bubbles. To counter polarization, it must ensure that clusters on topics like finance and healthcare include a diverse range of opinions. This is an example of a constrained clustering problem. The popular spectral clustering algorithm [Ng et al., 2001, von Luxburg, 2007] has been adapted over the years to include constraints such as *must-link* and *cannot-link* constraints [Kamvar et al., 2003, Wang and Davidson, 2010], size-balanced clusters [Banerjee and Ghosh, 2006], and statistical fairness [Kleindessner et al., 2019]. These constraints can be broadly divided into two categories: **(i)** *Population level* constraints that must be satisfied by the clusters as a whole (e.g. size-balanced clusters and statistical fairness); and **(ii)** *Individual level* constraints that must be satisfied at the level of individual nodes (e.g. must/cannot link constraints). To the best of our knowledge, the only known statistical consistency guarantees for constrained spectral clustering were studied in Kleindessner et al. [2019] in the context of a *population level* fairness constraint where the goal is to find clusters that are balanced with respect to a categorical sensitive node attribute. In this paper, we establish consistency guarantees for constrained spectral clustering under a new and more general *individual level* fairness constraint.

---

*Work done while the author was at the Indian Institute of Science, Bangalore.

36th Conference on Neural Information Processing Systems (NeurIPS 2022).

**Informal problem description:** We assume the availability of two graphs: a *similarity graph* $\mathcal{G}$ in which the clusters are to be found and a *representation graph* $\mathcal{R}$, defined on the same set of nodes as $\mathcal{G}$, which encodes the "is representative of" relationship. Our goal is to find clusters in $\mathcal{G}$ such that every node has a sufficient number of its representatives from $\mathcal{R}$ in all clusters. For example, $\mathcal{G}$ may be a graph of consumers based on the similarity of their purchasing habits and $\mathcal{R}$ may be a graph based on the similarity of their sensitive attributes such as gender, race, and sexual orientation. This, for instance, would then be a step towards reducing discrimination in online marketplaces [Fisman and Luca, 2016].

**Contributions and results:** *First*, in Section 3.1, we formalize our new individual level fairness constraint for clustering, called the *representation constraint*. It is different from most existing fairness notions which either apply at the population level [Chierichetti et al., 2017, Rösner and Schmidt, 2018, Bercea et al., 2019, Bera et al., 2019] or are hard to integrate with spectral clustering [Chen et al., 2019, Mahabadi and Vakilian, 2020, Anderson et al., 2020]. Unlike these notions, our constraint can be used with multiple sensitive attributes of different types (categorical, numerical etc.) and only requires observing an abstract representation graph based on these attributes rather than requiring their actual values, thereby discouraging individual profiling. Appendix A discusses the utility of individual fairness notions.

*Second*, in Section 3.2, we develop the *representation-aware* variant of unnormalized spectral clustering to find clusters that approximately satisfy the proposed constraint. An analogous variant for normalized spectral clustering is presented in Appendix B.2.

*Third*, in Section 4.1, we introduce $\mathcal{R}$-PP, a new representation-aware (or fair) planted partition model. This model generates random similarity graphs $\mathcal{G}$ conditioned on both the cluster membership of nodes and a given representation graph $\mathcal{R}$. Intuitively, $\mathcal{R}$-PP plants the properties of $\mathcal{R}$ in $\mathcal{G}$. We show that this model generates "hard" problem instances and establish the weak consistency[2] of our algorithms under this model for a class of $d$-regular representation graphs (Theorems 4.1 and 4.2). To the best of our knowledge, these are the first consistency results for constrained spectral clustering under an individual-level constraint. In fact, we show that our results imply the only other similar consistency result (but for a population-level constraint) in Kleindessner et al. [2019] as a special case (Appendix A).

Finally, *fourth*, we present empirical results on both real and simulated data to corroborate our theoretical findings (Section 5). In particular, our experiments show that our algorithms perform well in practice, even when the $d$-regularity assumption on $\mathcal{R}$ is violated.

**Related work:** Spectral clustering has been modified to satisfy individual level *must-link* and *cannot-link* constraints by pre-processing the similarity graph [Kamvar et al., 2003], post-processing the eigenvectors of the graph Laplacian [Li et al., 2009], and modifying its optimization problem [Yu and Shi, 2001, 2004, Wang and Davidson, 2010, Wang et al., 2014, Cucuringu et al., 2016]. It has also been extended to accommodate various population level constraints [Banerjee and Ghosh, 2006, Xu et al., 2009]. We are unaware of theoretical performance guarantees for any of these algorithms.

Of particular interest to us are the fairness constraints for clustering. One popular population level constraint requires sensitive attributes to be proportionally represented in clusters [Chierichetti et al., 2017, Rösner and Schmidt, 2018, Bercea et al., 2019, Bera et al., 2019, Esmaeili et al., 2020, 2021]. For example, if $50\%$ of the population is female then the same proportion should be respected in all clusters. Several efficient algorithms for discovering such clusters have been proposed [Schmidt et al., 2018, Ahmadian et al., 2019, Harb and Shan, 2020], though they almost exclusively focus on variants of $k$-means while we are interested in spectral clustering. Kleindessner et al. [2019] deserve a special mention as they develop a spectral clustering algorithm for this fairness notion. However, we recover all the results presented in Kleindessner et al. [2019] as a special case of our analysis as our proposed constraint interpolates between population level and individual level fairness based on the structure of $\mathcal{R}$. While individual fairness notions for clustering have also been explored [Chen et al., 2019, Mahabadi and Vakilian, 2020, Anderson et al., 2020, Chakrabarty and Negahbani, 2021], none of them have previously been used with spectral clustering. See Caton and Haas [2020] for a broader discussion on fairness.

---

[2]An algorithm is called weakly consistent if it makes $o(N)$ mistakes with probability $1 - o(1)$, where $N$ is the number of nodes in the similarity graph $\mathcal{G}$ [Abbe, 2018]

A final line of relevant work concerns consistency results for variants of unconstrained spectral clustering. von Luxburg et al. [2008] established the weak consistency of spectral clustering assuming that the similarity graph $\mathcal{G}$ encodes cosine similarity between examples using feature vectors drawn from a particular probability distribution. Rohe et al. [2011] and Lei and Rinaldo [2015] assume that $\mathcal{G}$ is sampled from variants of the Stochastic Block Model (SBM) [Holland et al., 1983]. Zhang et al. [2014] allow clusters to overlap. Binkiewicz et al. [2017] consider auxiliary node attributes, though, unlike us, their aim is to find clusters that are well *aligned* with these attributes. A faster variant of spectral clustering was analyzed by Tremblay et al. [2016]. Spectral clustering has also been studied on other types of graphs such as hypergraphs [Ghoshdastidar and Dukkipati, 2017a,b] and strong consistency guarantees are also known [Gao et al., 2017, Lei and Zhu, 2017, Vu, 2018], albeit under stronger assumptions.

**Notation:** Define $[n] := \{1, 2, \ldots, n\}$ for any integer $n$. Let $\mathcal{G} = (\mathcal{V}, \mathcal{E})$ denote a similarity graph, where $\mathcal{V} = \{v_1, v_2, \ldots, v_N\}$ is the set of $N$ nodes and $\mathcal{E} \subseteq \mathcal{V} \times \mathcal{V}$ is the set of edges. Clustering aims to partition the nodes in $\mathcal{G}$ into $K \geq 2$ non-overlapping clusters $\mathcal{C}_1, \ldots, \mathcal{C}_K \subseteq \mathcal{V}$. We assume the availability of another graph, called a *representation graph* $\mathcal{R} = (\mathcal{V}, \hat{\mathcal{E}})$, which is defined on the same set of vertices as $\mathcal{G}$ but with different edges $\hat{\mathcal{E}}$. The discovered clusters $\mathcal{C}_1, \ldots, \mathcal{C}_K$ are required to satisfy a fairness constraint encoded by $\mathcal{R}$, as described in Section 3.1. $\mathbf{A}, \mathbf{R} \in \{0, 1\}^{N \times N}$ denote the adjacency matrices of graphs $\mathcal{G}$ and $\mathcal{R}$, respectively. We assume that $\mathcal{G}$ and $\mathcal{R}$ are undirected and that $\mathcal{G}$ has no self-loops.

## 2 Unnormalized spectral clustering

We begin with a brief review of unnormalized spectral clustering which will be useful in describing our algorithm in Section 3.2. The normalized variants of traditional spectral clustering and our algorithm have been deferred to Appendix B. Given a similarity graph $\mathcal{G}$, unnormalized spectral clustering finds clusters by approximately minimizing the following metric known as ratio-cut [von Luxburg, 2007]

$$\text{RCut}(\mathcal{C}_1, \ldots, \mathcal{C}_K) = \sum_{i=1}^{K} \frac{\text{Cut}(\mathcal{C}_i, \mathcal{V} \backslash \mathcal{C}_i)}{|\mathcal{C}_i|}.$$

Here, $\mathcal{V} \backslash \mathcal{C}_i$ is the set difference between $\mathcal{V}$ and $\mathcal{C}_i$. For any two subsets $\mathcal{X}, \mathcal{Y} \subseteq \mathcal{V}$, $\text{Cut}(\mathcal{X}, \mathcal{Y}) = \frac{1}{2} \sum_{v_i \in \mathcal{X}, v_j \in \mathcal{Y}} A_{ij}$ counts the number of edges that have one endpoint in $\mathcal{X}$ and another in $\mathcal{Y}$. Let $\mathbf{D} \in \mathbb{R}^{N \times N}$ be a diagonal degree matrix where $D_{ii} = \sum_{j=1}^{N} A_{ij}$ for all $i \in [N]$. It is easy to verify that ratio-cut can be expressed in terms of the graph Laplacian $\mathbf{L} := \mathbf{D} - \mathbf{A}$ and a cluster membership matrix $\mathbf{H} \in \mathbb{R}^{N \times K}$ as $\text{RCut}(\mathcal{C}_1, \ldots, \mathcal{C}_K) = \text{trace}\{\mathbf{H}^\mathsf{T} \mathbf{L} \mathbf{H}\}$, where

$$H_{ij} = \begin{cases} \frac{1}{\sqrt{|\mathcal{C}_j|}} & \text{if } v_i \in \mathcal{C}_j \\ 0 & \text{otherwise.} \end{cases} \tag{1}$$

Thus, to find good clusters, one can minimize $\text{trace}\{\mathbf{H}^\mathsf{T} \mathbf{L} \mathbf{H}\}$ over all $\mathbf{H}$ that have the form given in (1). However, the combinatorial nature of this constraint makes this problem NP-hard [Wagner and Wagner, 1993]. Unnormalized spectral clustering instead solves the following relaxed problem:

$$\min_{\mathbf{H} \in \mathbb{R}^{N \times K}} \text{trace}\{\mathbf{H}^\mathsf{T} \mathbf{L} \mathbf{H}\} \quad \text{s.t.} \quad \mathbf{H}^\mathsf{T} \mathbf{H} = \mathbf{I}. \tag{2}$$

Note that $\mathbf{H}$ in (1) satisfies $\mathbf{H}^\mathsf{T} \mathbf{H} = \mathbf{I}$. The above relaxation is often referred to as the spectral relaxation. By Rayleigh-Ritz theorem [Lütkepohl, 1996, Section 5.2.2], the optimal matrix $\mathbf{H}^*$ is such that it has $\mathbf{u}_1, \mathbf{u}_2, \ldots, \mathbf{u}_K \in \mathbb{R}^N$ as its columns, where $\mathbf{u}_i$ is the eigenvector corresponding to the $i^{th}$ smallest eigenvalue of $\mathbf{L}$ for all $i \in [K]$. The algorithm clusters the rows of $\mathbf{H}^*$ into $K$ clusters using $k$-means clustering [Lloyd, 1982] to return $\hat{\mathcal{C}}_1, \ldots, \hat{\mathcal{C}}_K$. Algorithm 1 summarizes this procedure. Unless stated otherwise, we will use spectral clustering (without any qualification) to refer to unnormalized spectral clustering.

## 3 Representation constraint and representation-aware spectral clustering

In this section, we first describe our individual level fairness constraint in Section 3.1 and then develop *Unnormalized Representation-Aware Spectral Clustering* in Section 3.2 to find clusters that approximately satisfy this constraint. See Appendix B for the normalized variant of the algorithm.

---
**Algorithm 1** Unnormalized spectral clustering
---
1: **Input:** Adjacency matrix $\mathbf{A}$, number of clusters $K \geq 2$
2: Compute the Laplacian matrix $\mathbf{L} = \mathbf{D} - \mathbf{A}$.
3: Compute the first $K$ eigenvectors $\mathbf{u}_1, \ldots, \mathbf{u}_K$ of $\mathbf{L}$. Let $\mathbf{H}^* \in \mathbb{R}^{N \times K}$ be a matrix that has $\mathbf{u}_1, \ldots, \mathbf{u}_K$ as its columns.
4: Let $\mathbf{h}_i^*$ denote the $i^{th}$ row of $\mathbf{H}^*$. Cluster $\mathbf{h}_1^*, \ldots, \mathbf{h}_N^*$ into $K$ clusters using $k$-means clustering.
5: **Output:** Clusters $\hat{\mathcal{C}}_1, \ldots, \hat{\mathcal{C}}_K$, s.t. $\hat{\mathcal{C}}_i = \{v_j \in \mathcal{V} : \mathbf{h}_j^* \text{ was assigned to the } i^{th} \text{ cluster}\}$.
---

## 3.1 Representation constraint

A representation graph $\mathcal{R}$ connects nodes that represent each other based on sensitive attributes (e.g. political opinions). Let $\mathcal{N}_{\mathcal{R}}(i) = \{v_j : R_{ij} = 1\}$ be the set of neighbors of node $v_i$ in $\mathcal{R}$. The size of $\mathcal{N}_{\mathcal{R}}(i) \cap \mathcal{C}_k$ specifies node $v_i$'s representation in cluster $\mathcal{C}_k$. To motivate our constraint, consider the following notion of *balance* $\rho_i$ of clusters defined from the perspective of a particular node $v_i$:

$$\rho_i = \min_{k,\ell \in [K]} \frac{|\mathcal{C}_k \cap \mathcal{N}_{\mathcal{R}}(i)|}{|\mathcal{C}_\ell \cap \mathcal{N}_{\mathcal{R}}(i)|} \tag{3}$$

It is easy to see that $0 \leq \rho_i \leq 1$ and higher values of $\rho_i$ indicate that node $v_i$ has an adequate representation in all clusters. Thus, one objective could be to find clusters $\mathcal{C}_1, \ldots, \mathcal{C}_K$ that solve the following optimization problem.

$$\min_{\mathcal{C}_1, \ldots, \mathcal{C}_K} f(\mathcal{C}_1, \ldots, \mathcal{C}_K) \quad \text{s.t.} \quad \rho_i \geq \alpha, \ \forall \, i \in [N], \tag{4}$$

where $f(\cdot)$ is inversely proportional to the quality of clusters (such as RCut) and $\alpha \in [0, 1]$ is a user specified threshold. However, it is not clear how this approach can be combined with spectral clustering to develop a consistent algorithm. We take a different approach described below.

First, note that $\min_{i \in [N]} \rho_i \leq \min_{k,\ell \in [K]} \frac{|\mathcal{C}_k|}{|\mathcal{C}_\ell|}$. Therefore, the balance $\rho_i$ of the least balanced node $v_i$ is maximized when its representatives $\mathcal{N}_{\mathcal{R}}(i)$ are split across clusters $\mathcal{C}_1, \ldots, \mathcal{C}_K$ in proportion to their sizes. Representation constraint requires this condition to be satisfied for each node in the graph.

**Definition 3.1** (Representation constraint). *Given a representation graph $\mathcal{R}$, clusters $\mathcal{C}_1, \ldots, \mathcal{C}_K$ in $\mathcal{G}$ satisfy the representation constraint if $|\mathcal{C}_k \cap \mathcal{N}_{\mathcal{R}}(i)| \propto |\mathcal{C}_k|$ for all $i \in [N]$ and $k \in [K]$, i.e.,*

$$\frac{|\mathcal{C}_k \cap \mathcal{N}_{\mathcal{R}}(i)|}{|\mathcal{C}_k|} = \frac{|\mathcal{N}_{\mathcal{R}}(i)|}{N}, \ \ \forall k \in [K], \ \forall i \in [N]. \tag{5}$$

In other words, the representation constraint requires the representatives of any given node to have a proportional membership in all clusters. For example, if $v_i$ is connected to $30\%$ of all nodes in $\mathcal{R}$, then it must have $30\%$ representation in all clusters discovered in $\mathcal{G}$. It is important to note that this constraint applies at the level of individual nodes unlike population level constraints [Chierichetti et al., 2017].

While (4) can always be solved for a small enough value of $\alpha$ (with the convention that $0/0 = 1$), the constraint in Definition 3.1 may not always be feasible. For example, (5) can never be satisfied if a node has only two representatives (i.e., $|\mathcal{N}_{\mathcal{R}}(i)| = 2$) and there are $K > 2$ clusters. However, as exactly satisfying constraints in clustering problems is often NP-hard [Davidson and Ravi, 2005], most approaches look for approximate solutions. In the same spirit, our algorithms use spectral relaxation to approximately satisfy (5), ensuring their wide applicability even when exact satisfaction is impossible.

In practice, $\mathcal{R}$ can be obtained by computing similarity between nodes based on one or more sensitive attributes (say by taking $k$-nearest neighbors). These attributes can have different types as opposed to existing notions that expect categorical attributes (Appendix A). Moreover, once $\mathcal{R}$ has been calculated, the values of sensitive attributes need not be exposed to the algorithm, thus adding privacy. Appendix A presents a toy example to demonstrate the utility of individual level fairness and shows that (5) recovers the population level constraint from Chierichetti et al. [2017] and Kleindessner et al. [2019] for particular configurations of $\mathcal{R}$, thus recovering all results from Kleindessner et al. [2019] as a special case of our analysis.

Finally, while individual fairness notions have conventionally required similar individuals to be treated similarly [Dwork et al., 2012], our constraint requires similar individuals (neighbors in $\mathcal{R}$) to be spread across different clusters (Definition 3.1). This new type of individual fairness constraint may be of independent interest to the community. Next, we describe one of the proposed algorithms.

## 3.2 Unnormalized representation-aware spectral clustering (UREPSC)

The lemma below identifies a sufficient condition that implies the representation constraint and can be added to the optimization problem (2) solved by spectral clustering. See Appendix E for the proof.

**Lemma 3.1.** *Let $\mathbf{H} \in \mathbb{R}^{N \times K}$ have the form specified in* (1). *The condition*

$$\mathbf{R}\left(\mathbf{I} - \frac{1}{N}\mathbf{1}\mathbf{1}^\mathsf{T}\right)\mathbf{H} = \mathbf{0} \tag{6}$$

*implies that the corresponding clusters $\mathcal{C}_1, \ldots, \mathcal{C}_K$ satisfy the constraint in* (5). *Here, $\mathbf{I}$ is the $N \times N$ identity matrix and $\mathbf{1}$ is a $N$-dimensional all-ones vector.*

With the unnormalized graph Laplacian $\mathbf{L}$ defined in Section 2, we add the condition from Lemma 3.1 to the optimization problem after spectral relaxation in (2) and solve

$$\min_{\mathbf{H}} \quad \text{trace}\{\mathbf{H}^\mathsf{T}\mathbf{L}\mathbf{H}\} \quad \text{s.t.} \quad \mathbf{H}^\mathsf{T}\mathbf{H} = \mathbf{I}; \quad \mathbf{R}\left(\mathbf{I} - \frac{1}{N}\mathbf{1}\mathbf{1}^\mathsf{T}\right)\mathbf{H} = \mathbf{0}. \tag{7}$$

Clearly, the columns of any feasible $\mathbf{H}$ must belong to the null space of $\mathbf{R}(\mathbf{I} - \mathbf{1}\mathbf{1}^\mathsf{T}/N)$. Thus, any feasible $\mathbf{H}$ can be expressed as $\mathbf{H} = \mathbf{YZ}$ for some matrix $\mathbf{Z} \in \mathbb{R}^{N-r \times K}$, where $\mathbf{Y} \in \mathbb{R}^{N \times N-r}$ is an orthonormal matrix containing the basis vectors for the null space of $\mathbf{R}(\mathbf{I} - \mathbf{1}\mathbf{1}^\mathsf{T}/N)$ as its columns. Here, $r$ is the rank of $\mathbf{R}(\mathbf{I} - \mathbf{1}\mathbf{1}^\mathsf{T}/N)$. Because $\mathbf{Y}^\mathsf{T}\mathbf{Y} = \mathbf{I}$, $\mathbf{H}^\mathsf{T}\mathbf{H} = \mathbf{Z}^\mathsf{T}\mathbf{Y}^\mathsf{T}\mathbf{YZ} = \mathbf{Z}^\mathsf{T}\mathbf{Z}$. Thus, $\mathbf{H}^\mathsf{T}\mathbf{H} = \mathbf{I} \Leftrightarrow \mathbf{Z}^\mathsf{T}\mathbf{Z} = \mathbf{I}$. The following problem is equivalent to (7) by setting $\mathbf{H} = \mathbf{YZ}$.

$$\min_{\mathbf{Z}} \quad \text{trace}\{\mathbf{Z}^\mathsf{T}\mathbf{Y}^\mathsf{T}\mathbf{L}\mathbf{YZ}\} \quad \text{s.t.} \quad \mathbf{Z}^\mathsf{T}\mathbf{Z} = \mathbf{I}. \tag{8}$$

As in standard spectral clustering, the solution to (8) is given by the $K$ leading eigenvectors of $\mathbf{Y}^\mathsf{T}\mathbf{L}\mathbf{Y}$. Of course, for $K$ eigenvectors to exist, $N - r$ must be at least $K$ as $\mathbf{Y}^\mathsf{T}\mathbf{L}\mathbf{Y}$ has dimensions $N - r \times N - r$. The clusters can then be recovered by using $k$-means clustering to cluster the rows of $\mathbf{H} = \mathbf{YZ}$, as in Algorithm 1. Algorithm 2 summarizes this procedure. We refer to this algorithm as unnormalized representation-aware spectral clustering (UREPSC). We make three important remarks before proceeding with the theoretical analysis.

*Remark* 1 (Spectral relaxation). As $\mathbf{R}(\mathbf{I}-\mathbf{1}\mathbf{1}^\mathsf{T}/N)\mathbf{H} = \mathbf{0}$ implies the satisfaction of the representation constraint only when $\mathbf{H}$ has the form given in (1), a feasible solution to (7) may not necessarily result in *representation-aware* clusters. In fact, even in the unconstrained case, there are no general guarantees that bound the difference between the optimal solution of (2) and the original NP-hard ratio-cut problem [Kleindessner et al., 2019]. Thus, the representation-aware nature of the clusters discovered by solving (8) cannot be guaranteed in general (as is the case with [Kleindessner et al., 2019]). Nonetheless, we show in Section 4 that the discovered clusters indeed satisfy the constraint under certain additional assumptions.

*Remark* 2 (Computational complexity). Algorithm 2 has a time complexity of $O(N^3)$ and space complexity of $O(N^2)$. Finding the null space of $\mathbf{R}(\mathbf{I} - \mathbf{1}\mathbf{1}^\mathsf{T}/N)$ to calculate $\mathbf{Y}$ and computing the eigenvectors of appropriate matrices are the computationally dominant steps. This matches the worst-case complexity of Algorithm 1. For small $K$, several approximations can reduce this complexity, but most such techniques require $K = 2$ [Yu and Shi, 2004, Xu et al., 2009].

*Remark* 3 (Approximate UREPSC). Algorithm 2 requires $\text{rank}\{\mathbf{R}\} \leq N - K$ to ensure the existence of $K$ orthonormal eigenvectors of $\mathbf{Y}^\mathsf{T}\mathbf{L}\mathbf{Y}$. When a graph $\mathcal{R}$ violates this assumption, we instead use the best rank $R$ approximation of its adjacency matrix $\mathbf{R}$ ($R \leq N - K$) and refer to this algorithm as UREPSC (APPROX.). This approximation of $\mathbf{R}$ need not have binary elements, but it works well in practice (Section 5). Appendix C provides more intuition behind this low rank approximation, contrasts this strategy with clustering $\mathcal{R}$ to recover latent sensitive groups that can be reused with existing population level notions, and highlights the challenges associated with finding theoretical guarantees for UREPSC (APPROX.), which is an interesting direction for future work.

---

**Algorithm 2** UREPSC

---

1: **Input:** Adjacency matrix $\mathbf{A}$, representation graph $\mathbf{R}$, number of clusters $K \geq 2$
2: Compute $\mathbf{Y}$ containing orthonormal basis vectors of $\text{null}\{\mathbf{R}(\mathbf{I} - \frac{1}{N}\mathbf{1}\mathbf{1}^\intercal)\}$
3: Compute Laplacian $\mathbf{L} = \mathbf{D} - \mathbf{A}$
4: Compute leading $K$ eigenvectors of $\mathbf{Y}^\intercal\mathbf{L}\mathbf{Y}$. Let $\mathbf{Z}$ contain these vectors as its columns.
5: Apply $k$-means clustering to rows of $\mathbf{H} = \mathbf{Y}\mathbf{Z}$ to get clusters $\hat{\mathcal{C}}_1, \hat{\mathcal{C}}_2, \ldots, \hat{\mathcal{C}}_K$
6: **Return:** Clusters $\hat{\mathcal{C}}_1, \hat{\mathcal{C}}_2, \ldots, \hat{\mathcal{C}}_K$

---

## 4  Analysis

This section shows that Algorithms 2 and 4 (see Appendix B) recover ground truth clusters with high probability under certain assumptions on the representation graph. We begin by introducing the representation-aware planted partition model in Section 4.1.

### 4.1  $\mathcal{R}$-PP model

The well known Planted Partition random graph model independently connects two nodes in $\mathcal{V}$ with probability $p$ if they belong to the same cluster and $q$ otherwise, where the ground truth cluster memberships are specified by a function $\pi : \mathcal{V} \rightarrow [K]$. Below, we define a variant of this model *with respect to* a representation graph $\mathcal{R}$ and refer to it as the Representation-Aware (or Fair) Planted Partition model or $\mathcal{R}$-PP.

**Definition 4.1** ($\mathcal{R}$-PP). *A $\mathcal{R}$-PP is defined by the tuple $(\pi, \mathcal{R}, p, q, r, s)$, where $\pi : \mathcal{V} \rightarrow [K]$ maps nodes in $\mathcal{V}$ to clusters, $\mathcal{R}$ is a representation graph, and $1 \geq p \geq q \geq r \geq s \geq 0$ are probabilities used for sampling edges. Under this model, for all $i > j$,*

$$
\mathrm{P}(A_{ij} = 1) = \begin{cases} p & \text{if } \pi(v_i) = \pi(v_j) \text{ and } R_{ij} = 1, \\ q & \text{if } \pi(v_i) \neq \pi(v_j) \text{ and } R_{ij} = 1, \\ r & \text{if } \pi(v_i) = \pi(v_j) \text{ and } R_{ij} = 0, \\ s & \text{if } \pi(v_i) \neq \pi(v_j) \text{ and } R_{ij} = 0. \end{cases} \tag{9}
$$

Similarity graphs $\mathcal{G}$ sampled from $\mathcal{R}$-PP have two interesting properties: **(i)** Everything else being equal, nodes have a higher tendency of connecting with other nodes in the same cluster ($p \geq q$ and $r \geq s$); and **(ii)** Nodes connected in $\mathcal{R}$ have a higher probability of connecting in $\mathcal{G}$ ($p \geq r$ and $q \geq s$). Thus, $\mathcal{R}$-PP plants both the clusters in $\pi$ and the properties of $\mathcal{R}$ into the sampled graph $\mathcal{G}$.

*Remark* 4 ($\mathcal{R}$-PP and "hard" problem instances). Clusters satisfying (5) must proportionally distribute the nodes connected in $\mathcal{R}$ amongst themselves. However, $\mathcal{R}$-PP makes nodes connected in $\mathcal{R}$ more likely to connect in $\mathcal{G}$, even if they belong to different clusters ($q \geq r$). In this sense, graphs sampled from $\mathcal{R}$-PP are "hard" instances for our algorithms.

When $\mathcal{R}$ itself has latent groups, there are two natural ways to cluster the nodes: **(i)** Based on the clusters specified by $\pi$; and **(ii)** Based on the clusters in $\mathcal{R}$. The clusters based on option **(ii)** are likely to not satisfy (5) as tightly connected nodes in $\mathcal{R}$ will be assigned to the same cluster. We show in the next section that, under certain assumptions, $\pi$ can be defined so that the clusters encoded by it satisfy (5) by construction. Recovering these ground truth clusters (instead of other natural choices like option **(ii)**) then amounts to recovering *representation-aware* clusters.

### 4.2  Consistency results

As noted in Section 3.1, some representation graphs lead to constraints that cannot be satisfied. For our theoretical analysis, we restrict our focus to a case where the constraint in (5) is feasible. Towards this end, an additional assumption on $\mathcal{R}$ is required.

**Assumption 4.1.** *$\mathcal{R}$ is a $d$-regular graph for $K \leq d \leq N$. Moreover, $R_{ii} = 1$ for all $i \in [N]$ and each node in $\mathcal{R}$ is connected to $d/K$ nodes from cluster $\mathcal{C}_j$ for all $j \in [K]$ (including the self-loop).*

Assumption 4.1 ensures the existence of a $\pi$ for which the ground-truth clusters satisfy (5). Namely, assuming equal-sized clusters, set $\pi(v_i) = k$ if $(k-1)\frac{N}{K} \leq i \leq k\frac{N}{K}$ for all $i \in [N]$ and $k \in [K]$.

Before presenting our main results, we need additional notation. Let $\boldsymbol{\Theta} \in \{0,1\}^{N \times K}$ indicate the ground-truth cluster memberships encoded by $\pi$ (i.e., $\Theta_{ij} = 1 \Leftrightarrow v_i \in \mathcal{C}_j$) and $\hat{\boldsymbol{\Theta}} \in \{0,1\}^{N \times K}$ indicate the clusters returned by the algorithm ($\hat{\Theta}_{ij} = 1 \Leftrightarrow v_i \in \hat{\mathcal{C}}_j$). With $\mathcal{J}$ as the set of all $K \times K$ permutation matrices, the fraction of misclustered nodes is defined as $M(\boldsymbol{\Theta}, \hat{\boldsymbol{\Theta}}) = \min_{\mathbf{J} \in \mathcal{J}} \frac{1}{N} \|\boldsymbol{\Theta} - \hat{\boldsymbol{\Theta}}\mathbf{J}\|_0$ [Lei and Rinaldo, 2015]. Theorems 4.1 and 4.2 use the eigenvalues of the Laplacian matrix in the expected case, defined as $\mathcal{L} = \mathcal{D} - \mathcal{A}$, where $\mathcal{A} = \mathbb{E}[\mathbf{A}]$ is the expected adjacency matrix of a graph sampled from $\mathcal{R}$-PP and $\mathcal{D} \in \mathbb{R}^{N \times N}$ is its corresponding degree matrix. The next two results establish high-probability upper bounds on the fraction of misclustered nodes for UREPSC and NREPSC (see Appendix B) for similarity graphs $\mathcal{G}$ sampled from $\mathcal{R}$-PP.

**Theorem 4.1** (Error bound for UREPSC). *Let* $\operatorname{rank}\{\mathbf{R}\} \leq N - K$ *and assume that all clusters have equal sizes. Let* $\mu_1 \leq \mu_2 \leq \cdots \leq \mu_{N-r}$ *denote the eigenvalues of* $\mathbf{Y}^{\mathsf{T}}\mathcal{L}\mathbf{Y}$*, where* $\mathbf{Y}$ *was defined in Section 3.2. Define* $\gamma = \mu_{K+1} - \mu_K$*. Under Assumption 4.1, there exists a universal constant* $\operatorname{const}(C, \alpha)$*, such that if* $\gamma$ *satisfies* $\gamma^2 \geq \operatorname{const}(C, \alpha)(2 + \epsilon)pNK \ln N$ *and* $p \geq C \ln N / N$ *for some* $C > 0$*, then*

$$M(\boldsymbol{\Theta}, \hat{\boldsymbol{\Theta}}) \leq \operatorname{const}(C, \alpha) \frac{(2 + \epsilon)}{\gamma^2} pN \ln N$$

*for every* $\epsilon > 0$ *with probability at least* $1 - 2N^{-\alpha}$ *when a* $(1 + \epsilon)$*-approximate algorithm for k-means clustering is used in Step 5 of Algorithm 2.*

**Theorem 4.2** (Error bound for NREPSC). *Let* $\operatorname{rank}\{\mathbf{R}\} \leq N - K$ *and assume that all clusters have equal sizes. Let* $\mu_1 \leq \mu_2 \leq \cdots \leq \mu_{N-r}$ *denote the eigenvalues of* $\mathcal{Q}^{-1}\mathbf{Y}^{\mathsf{T}}\mathcal{L}\mathbf{Y}\mathcal{Q}^{-1}$*, where* $\mathcal{Q} = \sqrt{\mathbf{Y}^{\mathsf{T}}\mathcal{D}\mathbf{Y}}$ *and* $\mathbf{Y}$ *was defined in Section 3.2. Define* $\gamma = \mu_{K+1} - \mu_K$ *and* $\lambda_1 = qd + s(N - d) + (p - q)\frac{d}{K} + (r - s)\frac{N-d}{K}$*. Under Assumption 4.1, there are universal constants* $\operatorname{const}_1(C, \alpha)$*,* $\operatorname{const}_2(C, \alpha)$*, and* $\operatorname{const}_3(C, \alpha)$ *such that if:*

1. $\left(\frac{\sqrt{pN \ln N}}{\lambda_1 - p}\right)\left(\frac{\sqrt{pN \ln N}}{\lambda_1 - p} + \frac{1}{6\sqrt{C}}\right) \leq \frac{1}{16(\alpha + 1)}$,

2. $\frac{\sqrt{pN \ln N}}{\lambda_1 - p} \leq \operatorname{const}_2(C, \alpha)$*, and*

3. $16(2 + \epsilon)\left[\frac{8\operatorname{const}_3(C, \alpha)\sqrt{K}}{\gamma} + \operatorname{const}_1(C, \alpha)\right]^2 \frac{pN^2 \ln N}{(\lambda_1 - p)^2} < \frac{N}{K}$,

*and* $p \geq C \ln N / N$ *for some* $C > 0$*, then,*

$$M(\boldsymbol{\Theta}, \hat{\boldsymbol{\Theta}}) \leq 32(2 + \epsilon)\left[\frac{8\operatorname{const}_3(C, \alpha)\sqrt{K}}{\gamma} + \operatorname{const}_1(C, \alpha)\right]^2 \frac{pN \ln N}{(\lambda_1 - p)^2},$$

*for every* $\epsilon > 0$ *with probability at least* $1 - 2N^{-\alpha}$ *when a* $(1 + \epsilon)$*-approximate algorithm for k-means clustering is used in Step 6 of Algorithm 4.*

All proofs have been deferred to Appendix D. Briefly, we show that the top $K$ eigenvectors of $\mathcal{L}$ **(i)** recover ground-truth clusters in the expected case (Lemmas D.1 to D.3) and **(ii)** lie in the null space of $\mathbf{R}(\mathbf{I} - \mathbf{1}\mathbf{1}^{\mathsf{T}}/N)$ and hence are also the top $K$ eigenvectors of $\mathbf{Y}^{\mathsf{T}}\mathcal{L}\mathbf{Y}$ (Lemma D.4). Matrix perturbation arguments then establish a high probability mistake bound in the general case when the graph $\mathcal{G}$ is sampled from a $\mathcal{R}$-PP (Lemmas D.5–D.8). Next, we discuss our assumptions and use the error bounds above to establish the weak consistency of our algorithms.

### 4.3 Discussion

Note that $\mathbf{I} - \mathbf{1}\mathbf{1}^{\mathsf{T}}/N$ is a projection matrix and $\mathbf{1}$ is its eigenvector with eigenvalue 0. Any vector orthogonal to $\mathbf{1}$ is an eigenvector with eigenvalue 1. Thus, $\operatorname{rank}\{\mathbf{I} - \mathbf{1}\mathbf{1}^{\mathsf{T}}/N\} = N - 1$. Because $\operatorname{rank}\{\mathbf{R}(\mathbf{I} - \mathbf{1}\mathbf{1}^{\mathsf{T}}/N)\} \leq \min(\operatorname{rank}\{\mathbf{R}\}, \operatorname{rank}\{\mathbf{I} - \mathbf{1}\mathbf{1}^{\mathsf{T}}/N\})$, requiring $\operatorname{rank}\{\mathbf{R}\} \leq N - K$ ensures that $\operatorname{rank}\{\mathbf{R}(\mathbf{I} - \mathbf{1}\mathbf{1}^{\mathsf{T}}/N)\} \leq N - K$, which is necessary for (8) to have a solution. The assumption on the size of the clusters and the $d$-regularity of $\mathcal{R}$ allows us to compute the smallest $K$ eigenvalues of the Laplacian matrix in the expected case. This is a crucial step in our proof.

In Theorem 4.2, $\lambda_1$ is defined such that the largest eigenvalue of the expected adjacency matrix under the $\mathcal{R}$-PP model is given by $\lambda_1 - p$ (see (15) and Lemma D.2). The three assumptions in

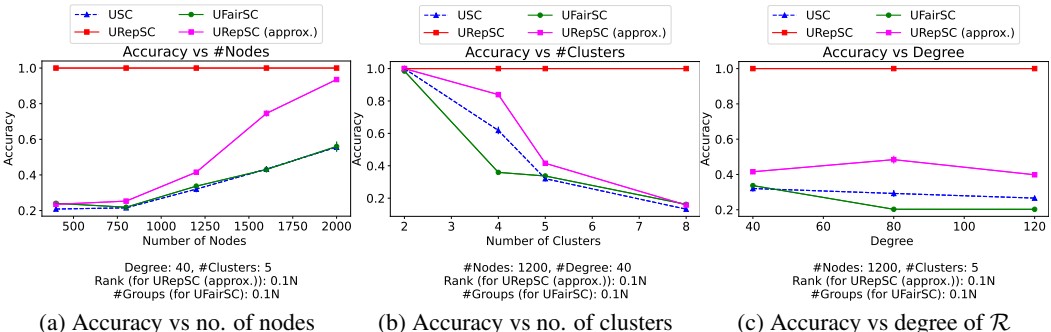

(a) Accuracy vs no. of nodes     (b) Accuracy vs no. of clusters     (c) Accuracy vs degree of $\mathcal{R}$

Figure 1: Comparing UREPSC with other "unnormalized" algorithms using synthetically generated $d$-regular representation graphs.

Theorem 4.2 essentially control the minimum rate at which $\lambda_1$ must grow with $N$. For instance, the first two assumptions can be replaced by a simpler but slightly stronger (yet practical) condition: $\lambda_1 - p = \omega(\sqrt{pN \ln N})$. This, for example, is satisfied in a realistic setting where $K = O(\ln N)$ and $d = O(\ln N)$. The third assumption also controls the rate of growth of $\lambda_1$, but in the context of the community structure contained in $\mathcal{Q}^{-1}\mathbf{Y}^T\mathcal{L}\mathbf{Y}\mathcal{Q}^{-1}$, as encoded by the eigengap $\gamma$. So, $\lambda_1$ must not increase at the expense of the community structure (e.g., by setting $p = q = r = s = 1$).

*Remark* 5. In practice, Algorithms 2 and 4 only require the rank assumption on $\mathbf{R}$ to ensure the feasibility of the corresponding optimization problems. The assumptions on the size of clusters and $d$-regularity of $\mathcal{R}$ are only needed for our theoretical analysis.

The next two corollaries establish the weak consistency of our algorithms as a direct consequence of Theorems 4.1 and 4.2.

**Corollary 4.1** (Weak consistency of UREPSC). *Under the same setup as Theorem 4.1, for* UREPSC, $M(\mathbf{\Theta}, \hat{\mathbf{\Theta}}) = o(1)$ *with probability* $1 - o(1)$ *if* $\gamma = \omega(\sqrt{pNK \ln N})$.

**Corollary 4.2** (Weak consistency of NREPSC). *Under the same setup as Theorem 4.2, for* NREPSC, $M(\mathbf{\Theta}, \hat{\mathbf{\Theta}}) = o(1)$ *with probability* $1 - o(1)$ *if* $\gamma = \omega(\sqrt{pNK \ln N}/(\lambda_1 - p))$.

The conditions on $\gamma$ are satisfied in many interesting cases. For example, when there are $P$ *protected groups* as in Chierichetti et al. [2017], the equivalent representation graph has $P$ cliques that are not connected to each other (see Appendix A). Kleindessner et al. [2019] show that $\gamma = \theta(N/K)$ in this case (for the unnormalized variant), which satisfies the criterion given above if $K$ is not too large.

Finally, Theorems 4.1 and 4.2 require a $(1 + \epsilon)$-approximate solution to $k$-means clustering. Several efficient algorithms have been proposed in the literature for this task [Kumar et al., 2004, Arthur and Vassilvitskii, 2007, Ahmadian et al., 2017]. Such algorithms are also available in commonly used software packages like MATLAB and scikit-learn[3]. The assumption that $p \geq C \ln N/N$ controls the sparsity of the graph and is required in the consistency proofs for standard spectral clustering as well [Lei and Rinaldo, 2015].

## 5 Numerical results

We experiment with three types of graphs: synthetically generated $d$-regular and non-$d$-regular representation graphs and a real-world dataset. See Appendix F for analogous results for NREPSC, the normalized variant of our algorithm. Before proceeding further, we make an important remark.

*Remark* 6 (Comparison with Kleindessner et al. [2019]). We refer to the unnormalized variant of the algorithm in Kleindessner et al. [2019] as UFAIRSC. It assumes that each node belongs to one of the $P$ protected groups $\mathcal{P}_1, \ldots, \mathcal{P}_P \subseteq \mathcal{V}$ that are observed by the learner. UREPSC recovers UFAIRSC as a special case when $\mathcal{R}$ is block diagonal (Appendix A). To demonstrate the generality of UREPSC,

---

[3]The algorithm in Kumar et al. [2004] runs in linear time in $N$ only when $K$ is a constant. When $K$ grows with $N$, one can instead use other practical variants whose average time complexity is linear in both $N$ and $K$ (e.g., in scikit-learn). These variants are often run with multiple seeds in practice to avoid local minima.

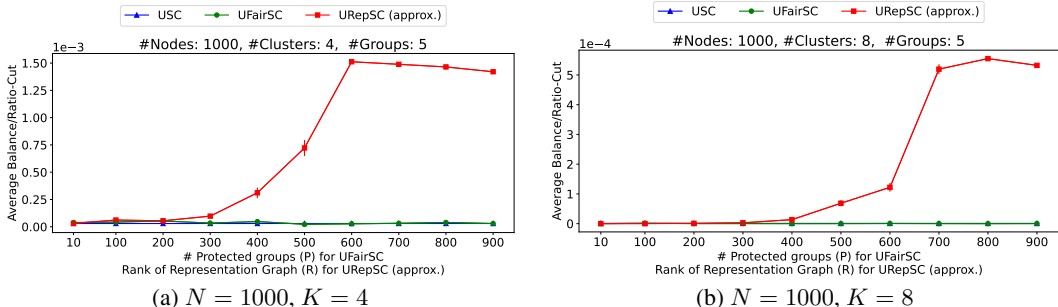

(a) $N = 1000, K = 4$        (b) $N = 1000, K = 8$

Figure 2: Comparing UREPSC (APPROX.) with UFAIRSC using synthetically generated representation graphs sampled from an SBM.

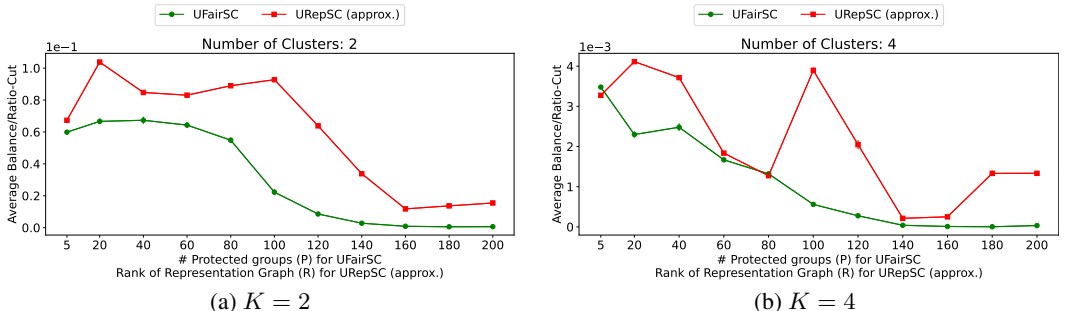

(a) $K = 2$        (b) $K = 4$

Figure 3: Comparing UREPSC (APPROX.) with UFAIRSC on FAO trade network.

we only experiment with $\mathcal{R}$'s that are not block diagonal. As UFAIRSC is not directly applicable in this setting, to compare with it, we approximate the protected groups by clustering the nodes in $\mathcal{R}$ using standard spectral clustering. Each discovered cluster is then treated as a protected group (also see Appendix C).

**$d$-regular representation graphs:** We sampled similarity graphs from $\mathcal{R}$-PP using $p = 0.4$, $q = 0.3$, $r = 0.2$, and $s = 0.1$ for various values of $d$, $N$, and $K$, while ensuring that the underlying $\mathcal{R}$ satisfies Assumption 4.1 and $\text{rank}\{\mathbf{R}\} \leq N - k$. Figure 1 compares UREPSC with unnormalized spectral clustering (USC) (Algorithm 1) and UFAIRSC. Figure 1a shows the effect of varying $N$ for a fixed $d = 40$ and $K = 5$. Figure 1b varies $K$ and keeps $N = 1200$ and $d = 40$ fixed. Similarly, Figure 1c keeps $N = 1200$ and $K = 5$ fixed and varies $d$. In all cases, we use $R = P = N/10$. The figures plot the accuracy on $y$-axis and report the mean and standard deviation across 10 independent executions. As the ground truth clusters satisfy Definition 3.1 by construction, a high accuracy implies that the algorithm returns representation-aware clusters. In Figure 1a, it appears that even USC will return representation-aware clusters for a large enough graph. However, this is not true if the number of clusters increases with $N$ (Figure 1b), as is common in practice.

**Representation graphs sampled from planted partition model:** We divide the nodes into $P = 5$ protected groups and sample a representation graph $\mathcal{R}$ using a (traditional) planted partition model with within (resp. across) protected group(s) connection probability given by $p_{\text{in}} = 0.8$ (resp. $p_{\text{out}} = 0.2$). Conditioned on $\mathcal{R}$, we then sample similarity graphs from $\mathcal{R}$-PP using the same parameters as before. We only experiment with UREPSC (APPROX.) as an $\mathcal{R}$ generated this way may violate the rank assumption. Moreover, because such an $\mathcal{R}$ may not be $d$-regular, high accuracy no longer implies representation awareness, and we instead report the ratio of average individual balance $\bar{\rho} = \frac{1}{N} \sum_{i=1}^{N} \rho_i$ (see (3)) to the value of $\text{RCut}$ in Figure 2. Higher values indicate high quality clusters that are also balanced from the perspective of individuals. Figure 2 shows a trade-off between accuracy and representation awareness. One can choose an appropriate value of $R$ in UREPSC (APPROX.) to get good quality clusters with a high balance.

**A real-world network:** The FAO trade network from United Nations is a multiplex network with 214 nodes representing countries and 364 layers representing commodities like coffee and barley [Domenico et al., 2015]. Edges corresponding to the volume of trade between countries. We connect each node to its five nearest neighbors in each layer and make the edges undirected. The first 182 layers are aggregated to construct $\mathcal{R}$ (connecting nodes that are connected in at least one layer) and the next 182 layers are used for constructing $\mathcal{G}$. Note that $\mathcal{R}$ constructed this way is not $d$-regular. To motivate this experiment further, note that clusters based only on $\mathcal{G}$ only consider the trade of commodities 183–364. However, countries also have other trade relations in $\mathcal{R}$, leading to shared economic interests. If the members of each cluster jointly formulate their economic policies, countries have an incentive to influence the economic policies of all clusters by having their representatives in them.

As before, we use the low-rank approximation for the representation graph in UREPSC (APPROX.). Figure 3 compares UREPSC (APPROX.) with UFAIRSC and has the same semantics as Figure 2. Different plots in Figure 3 correspond to different choices of $K$. UREPSC (APPROX.) achieves a higher ratio of average balance to ratio-cut. In practice, a user would choose $R$ by assessing the relative importance of a quality metric like ratio-cut and representation metric like average balance.

Appendix F contains results for NREPSC, experiments with another real-world network, a few additional experiments related to UREPSC, and a numerical validation of our time-complexity analysis. Appendix G contains plots that show both average balance and ratio-cut instead of their ratio.

## 6   Conclusion

We studied the consistency of constrained spectral clustering under a new individual level fairness constraint, called the representation constraint, using a novel $\mathcal{R}$-PP random graph model. Our work naturally generalizes existing population level constraints [Chierichetti et al., 2017] and associated spectral algorithms [Kleindessner et al., 2019]. Four important avenues for future work include **(i)** the relaxation of the $d$-regularity assumption in our analysis (needed to ensure representation awareness of ground-truth clusters), **(ii)** better theoretical understanding of UREPSC (APPROX.), **(iii)** improvement of the computational complexity of our algorithms, and **(iv)** exploring relaxed variants of our constraint and other (possibly non-spectral) algorithms for finding representation-aware clusters under such a relaxed constraint.

## Acknowledgments and Disclosure of Funding

The authors would like to thank the Science and Engineering Research Board (SERB), Department of Science and Technology, Government of India, for their generous funding towards this work through the IMPRINT project: IMP/2019/000383. The authors also thank Muni Sreenivas Pydi for reviewing the manuscript and providing valuable suggestions.

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
