# OpenReview forum: "Consistency of Constrained Spectral Clustering under Graph Induced Fair Planted Partitions"
_NeurIPS.cc/2022/Conference — NeurIPS 2022 Accept_

### Official Review · Reviewer_N1aR · 2022-07-04

**Rating:** 7
**Confidence:** 3
**Soundness:** 4 excellent
**Presentation:** 4 excellent
**Contribution:** 3 good

**Summary:**

This paper studies the problem of fairness in the spectral clustering algorithm. In particular, the goal is to design an algorithm which guarantees that nodes with 'similar' latent properties are evenly distributed across the clusters.

To achieve this, the paper introduces a new notion of fairness, with respect to a 'representation graph' which links nodes with shared properties. Then, it describes a new, fair spectral clustering algorithm which takes as input both the observed graph, and the representation graph, and returns a fair clustering.
The paper also introduces "a representation-aware planted partition model" which is a generalization of the classical planted partition model to take into account the representation graph.

Finally, the paper provides theoretical guarantees that the new algorithm successfully recovers the correct clusters in the planted partition model, under some reasonable assumptions.

**Questions:**

Can this framework can be used to control for multiple sensitive characteristics simultaneously (such as race *and* gender)? As I understand it, this couldn't be captured by the representation graph. This could be an interesting question for future work.

**Limitations:**

The authors have adequately addressed the limitations of their work.

**Strengths And Weaknesses:**

Strengths
----------
1. The paper is well written, well structured, and well motivated.
2. The paper studies an interesting and relevant problem.
3. The 'representation graph' is a novel contribution which naturally generalizes previous notions of fairness and may find uses in other applications.

Weaknesses
---------------
Minor issues:
1. The assumptions on the parameters in Theorem 4.2 are a little difficult to interpret. It would be useful to provide some more discussion with some intuition for how they should be interpreted.
2. Theorem 4.2 uses $\mathrm{const}_1$, $\mathrm{const}_4$, and $\mathrm{const}_5$. Should this instead be $1$, $2$, and $3$?

---

> ### Author Response · Authors · 2022-08-01
> **Regarding Theorem 4.2 and the use of multiple sensitive attributes**
>
> **Update 1:** All the changes mentioned below have now been added to the revised manuscript. Please look for the changes highlighted in brick-red color.
>
> Thank you for an overall positive review of our paper and for appreciating our new notion of fairness. We address your concerns below.
>
> 1. **Regarding Theorem 4.2 [answer shared with Reviewer 1o3a]:**
> We begin with $\lambda_1$. $\lambda_1$ is the largest eigenvalue of a slightly modified expected adjacency matrix under the $\mathcal{R}$-PP model (the largest eigenvalue of the expected adjacency matrix is given by $\lambda_1 - p$ (see eq. (15) and Lemma D.2)).
> All three assumptions in Theorem 4.2 essentially control the minimum rate at which $\lambda_1$ must grow with $N$. For instance, the first two assumptions can be replaced by a simpler but slightly stronger (yet practical) condition: $\lambda_1 - p = \omega(\sqrt{p N \ln N})$. This, for example, is satisfied when $K = O(\ln N)$ and $d = O(\ln N)$ which is not unrealistic.
> The third assumption is more involved because of the presence of the eigengap $\gamma$. While this assumption too controls the rate of growth of $\lambda_1$, it does so in context of the community structure contained in $\mathcal{Q}^{-1} \mathbf{Y}^T \mathcal{L} \mathbf{Y} \mathcal{Q}^{-1}$, as encoded by the eigengap $\gamma$ (please see Theorem 4.2 for the definition of these symbols). So while $\lambda_1$  must increase, it must not be at the expense of the community structure (for instance, when p=q=r=s=1).
> We will use the extra space in the camera ready version to add this discussion to the paper.
> Finally, the constants in Theorem 4.2 have been indexed in the order in which they appear in our proof. $\mathrm{const}_2$ and $\mathrm{const}_3$ were only used in intermediate lemmas and hence do not appear in Theorem 4.2. We will reindex these constants to avoid confusion.
>
> 2. **Regarding multiple sensitive attributes and multiple representation graphs [answer shared with Reviewer yXMv]**
> This is an interesting point. First, we would like to stress again that we can indeed construct a single $\mathcal{R}$ using multiple sensitive attributes by, for instance, using nearest neighbors based on an appropriate kernel (also see L147-148 and the comment from Reviewer yXMv).
> Second, the case where we have multiple $\mathcal{R}$ is more challenging. Among other things, these representation graphs may pose conflicting representation requirements. A simple workaround would be to construct a single $\mathcal{R}$ which is the “union” of all given representation graphs and then use it in our current algorithm. We have however not explored this setting and leave it for future work. Thanks for the suggestion.

---

> > ### Author Response · Authors · 2022-08-08
> > **Regarding our replies to questions raised by Reviewer N1aR**
> >
> > We have given detailed responses to several important questions raised by Reviewer N1aR. We request the reviewer to let us know if they have any further questions. We will be happy to provide answers. This will really help us to improve the paper. Thanks again.

---

### Official Review · Reviewer_yXMv · 2022-07-08

**Rating:** 6
**Confidence:** 4
**Soundness:** 3 good
**Presentation:** 3 good
**Contribution:** 2 fair

**Summary:**

The paper introduces an extension to graph clustering where each node needs to be "represented" in each cluster roughly proportionally to the size of each cluster. After defining the unnormalized (and normalized in appendix B) representation aware spectral clustering, proposes a modification to the spectral algorithm so that representation is maximized. After providing some in depth analysis of their algorithm in the planted partition, the authors present numerical results in synthetic data and one real network.

**Questions:**

- For spectral K-clustering another objective is using the max instead of the sum (Khandekar, R., Kortsarz, G., and Mirrokni, V. On the advantage of overlapping clusters for minimizing conductance. Algorithmica, 69(4):844–863, 2014.) Have you also considered this objective?
- Could the error bounds in section 4 incorporate p, q, r, s similarly to [Abbe, Emmanuel and Bandeira, Afonso S and Hall, Georgina, Exact Recovery in the Stochastic Block Model]?
- Is $\mathcal{R}$ in 4.1 a uniformly random graph? With the difference that the connections in each cluster are *exactly* $d/K$ instead of being on average $d/K$. This would make $\mathbb{R}$ to be approximately rank-1.
- Why are some xlabels missing eg 300 in fig 2, 900 in fig 6d, and more?
- What is this spike in fig 3b at rank=100 for URespSC (approx.)?
- Why did the authors decide to combine average balance and Ratio-Cut? I would like to see both separately.
- Representation needs to be a single graph, potentially aggregating a number of features. Could this be extended to multiple attributes (eg age, gender, income)?
- On commercial hardware, to what sizes of graphs does this method scale?
- According to [Leskovec, J., Lang, K. J., Dasgupta, A., and Mahoney, M. W. Community structure in large networks: Natural cluster sizes and the absence of large well-defined clusters. Internet Mathematics, 6(1):29–123, 2009] most interaction graphs have a very different structure than SBM. How do you expect your method to perform in such graphs?

**Limitations:**

- The theoretical guarantees are only for a very well-tailored planted partition model. This has been acknowledged by the authors as a limitation.
- Authors have acknowledged their method's computational bottleneck.
- Authors are up front about the scope of their analysis and have set goals to expand it.

**Strengths And Weaknesses:**

Strengths:
- The paper provides a new problem that is more expressive than previous work (originality, significance).
- For the $\mathcal{R}$-PP model the authors provide an in-depth analysis (quality).
- The paper is well structured and easily understood with the correct chapters relegated to the appendix, making it an enjoyable read (clarity).
- The appendix contains detailed proofs of the authors' claims and theorems.
- The authors consider both unnormalized ${\bf and}$ normalized spectral clustering.
- The algorithm that the authors provide is simple and comes with guarantees in certain cases.

Weaknesses:
- The authors define a very interesting problem in equation 4 and then immediately abandon it for a different problem in equation 5 two paragraphs later. Remove, or move to the appendix.
- The results section consists of describing the experimental setup, placing the plots, and never commenting on them.
- The numerical results contain only one real network whose quality is questionable. It's not necessarily bad, but it's an arbitrary construction.
- The authors do not present any numerical results for the time scaling of their methods (though theoretical analysis is present).

---

> ### Author Response · Authors · 2022-08-01
> **Regarding our experiments, assumptions, alternative formulations, and other issues**
>
> **Update 1:** All the changes mentioned below have now been added to the revised manuscript. Please look for the changes highlighted in brick-red color.
>
> Thank you for your thoughtful comments and astute questions. Hopefully our responses below adequately address your concerns.
>
> 1. **A note on Equation (4):**
> As you have pointed out, Equation (4) formulates an interesting problem. While we only use it as a stepping stone to Equation (5), which can be solved using spectral methods, we believe that this problem formulation may be of independent interest in general (as also hinted by Reviewer 7uut) and hence we prefer keeping it in the paper.
>
> 2. **Regarding experiments:**
> Please also see our common response titled “Additional plots” for an additional real-data experiment, a run-time plot, and separate plots for ratio-cut and average balance. We will add all of this information to the supplementary material in the revision.
> These experiments were executed on a Macbook Pro with an 8 core Intel i9 processor and 32 GB memory. The run-time plot should therefore provide an adequate idea about the size of the graphs that can be easily handled.
> Regarding missing x-labels in some figures, thanks for pointing it out. We have verified this again: the results reported are correct and we will fix x-axis values in the revision.
> Finally, the spike in Figure 3(b) appears due to a spike in average balance for that value of the rank $R$ used to approximate $\mathcal{R}$. Getting a theoretical understanding of why this happens is a challenging problem (see Appendix C for details). As we mention in L334-335, in practice, we expect the users to choose $R$ by looking at plots similar to Figure 3(b).
>
> 3. **Considering max instead of sum:**
> We believe that the reviewer is referring to the problem of finding overlapping clusters. We certainly agree that it is an interesting problem. Besides being practically relevant, overlapping clusters also pose interesting questions from the perspective of fairness. For instance, under our notion, an individual can have the same representative serving in two clusters simultaneously. The scope of our paper is restricted to the traditional non-overlapping clustering objective but we would be happy to study this problem in a future work (a good starting point would be the approach in Zhang et al. 2014). Thanks for the suggestion.
>
> 4. **Regarding $d$-regularity of $\mathcal{R}$:**
> We refer you to our answer to Reviewer 7uut titled “Regarding Assumption 4.1” if your question is about the need for exact $d/K$ connections in our analysis.
> However, there is another interesting interpretation here. When clusters are equally sized and $\mathcal{R}$ is $d$-regular, what if we only want to enforce $d/K$ representatives of each node in each cluster on an average? This is a weaker objective and, in our opinion, is not perfectly aligned with the aim of “individual” fairness. Certain nodes may still end up being over-represented in certain clusters while other nodes have very few representatives. Having said that, this weaker notion may be enough in certain applications and is definitely worth exploring.
>
> 5. **Can our error bounds be in terms of $p$, $q$, $r$, and $s$?**
> We would like to note that $\gamma$ in Theorems 4.1 and 4.2 and $\lambda_1$ in Theorem 4.2, do depend on $p$, $q$, $r$, and $s$. Unfortunately, a more interpretable dependence on $p$, $q$, $r$, and $s$ evades our analysis, as is the case even for unconstrained spectral clustering (for example, see Theorem 3.1 in Rohe et al. 2011 or Theorem 3.1 in Lei and Rinaldo 2015).
>
> 6. **Using multiple sensitive attributes and multiple representation graphs [answer shared with Reviewer N1aR]:**
> This is an interesting point. First, as you have correctly noted, we can construct a single $\mathcal{R}$ using multiple sensitive attributes by, for instance, using nearest neighbors based on an appropriate kernel.
> Second, the case where we have multiple $\mathcal{R}$ is more challenging. Among other things, these representation graphs may pose conflicting representation requirements. A simple workaround would be to construct a single $\mathcal{R}$ which is the “union” of all given representation graphs and then use it in our current algorithm. We have however not explored this setting and leave it for future work. Thanks for the suggestion.
>
> 7. **Real-world networks are different from SBM:**
> We experimentally demonstrate in Section 5 that our method works well for real world networks, even when they deviate from SBMs. The same holds true for standard spectral clustering which has been extensively analyzed under variants of SBMs (Rohe et al. 2011; Zhang et al. 2014; Lei and Rinaldo 2015; Binkiewicz et al. 2017), while also being one of the most popular clustering algorithms in practice. Analyzing our algorithms under more complex network models (such as degree-corrected SBMs) is an interesting direction for future work.

---

> > ### Comment · Reviewer_yXMv · 2022-08-04
> > **Response**
> >
> > I would like to thank the authors for their thoughtful comments.
> > 1. I agree that it is a problem of independent interest. I respect the authors decision to leave it in.
> > 2. The peak I mentioned is just an example of things I expect to be commented on when such plots are present. I am sure that the authors will have a more extensive discussion on the results of their experiments.
> > 3. This is not overlapping clustering. While it is interesting, I was referring to the alternative where the objective is not to minimize the summation of conductances, but the maximum conductance among the clusters. This is the difference between L-$1$ and L-$\infty$ norms. In your paper you are using the L-$1$. The difference is not very important in practice.
> > 4. No comments
> > 5. I understand and sympathize. Would be very interesting, it is also quite challenging.
> > 6. Leave as future work.
> > 7. Working on one or two real world datasets is not in my opinion exhaustive. Especially the UN dataset that has been turned into a graph by the authors in a random way. However, I understand that this paper's contributions go beyond practical applications.
> >
> > In conclusion, I believe this to be a well written paper regarding a niche extension of a well known problem and a positive addition to the published corpus. I recommend the acceptance of this submission and look forward to further work by the authors.

---

> > > ### Author Response · Authors · 2022-08-05
> > > **Thank you for your response**
> > >
> > > Thank you for the response and for clarifying the point regarding sum vs max in the objective function.
> > >
> > > Looking through the lens of fairness, our first thought is that using $\ell_{\infty}$ norm (minimizing the maximum ratio-cut across clusters) is already a step towards fairness (albeit under a different notion) because this objective does not allow many clusters to become "good" at the expense of making one cluster "bad". We have not considered this objective.
> > >
> > > Our second thought is that the optimisation problem in Equation (4) does something similar, but for balance and not ratio-cut. That is, it requires the minimum balance to be greater than some $\alpha$. However, this does not prevent different individuals from having different balances unless $\alpha$ is large enough. Again, this may be something worth exploring further.
> > >
> > > We sincerely thank you for the pointer. It will help us in shaping the direction of our future works.
> > >
> > > Thank you also for recommending an acceptance.

---

> > > > ### Author Response · Authors · 2022-08-06
> > > > **Regarding the scores**
> > > >
> > > > We would like to draw your attention to your (reviewer yXMv) comment: " I recommend the acceptance of this submission...". Thanks again. In this regard, we request you to consider increasing your score as you deem fit.

---

### Official Review · Reviewer_1o3a · 2022-07-11

**Rating:** 6
**Confidence:** 4
**Soundness:** 3 good
**Presentation:** 4 excellent
**Contribution:** 3 good

**Summary:**

This paper studies the use of spectral clustering under a setting where there is an underlying auxiliary _representation_ graph on the vertices, which encodes a notion of "fairness/balancedness" in the clusters. The goal under this setting is to find clusters, such that each clusters contains a representative number of vertices with respect to both the sparsest cut/conductance, and also with respect to the representation graph. To study this new setting, the paper introduces a new objective function, planted partition model, and the paper also performs experiments on synthetic and a real-world dataset.

**Questions:**

- What is the comparison between different methods with regards to the ratio cut objective? Is there a tradeoff between ensuring representation fairness and the ratio cut? It would be interesting to see if there is a way to maximize both, without losing performance in either of the objectives.

- How should a reader interpret $\lambda_1$, introduced in Theorem 4.2?

**Limitations:**

yes

**Strengths And Weaknesses:**

The strength of this paper is the new setting under which it proposes to study spectral clustering. Constrained/fair clustering has received considerable attention in recent years in the machine learning community, however not much work has been done in the Spectral Clustering literature, especially in the setting that this paper proposes (similar nodes should be spread across all clusters). In section 3, the paper clearly and convincingly introduces the new objectives under representation constraints, which seem natural to consider, and then develops a sensible algorithm to optimize these objectives. Guarantees are provided in Section 4, which seem correct to my knowledge. Experimental results validate the strength of the proposed method on both synthetic and real-world data.

Moreover, the paper is well-written and has a clear exposition.

A minor/medium weakness of the paper is in the theoretical analysis. Although the analysis is well-done, it relies on quite strong assumptions on the representation graph (d-regularity, balanced size clusters). Especially given the fact that in practice a representation graph might be constructed by hand, these conditions from Assumption 4.1 would be hard to satisfy in practice, making the theoretical analysis less applicable.

Another medium weakness of this paper is the presentation of Theorem 4.2
Although Theorem 4.1 has clear and interpretable results, theorem 4.2 is hard to parse, and it is difficult for me to understand whether the 3 conditions are easy to satisfy for example. There are many variables and constants defined ($\lambda_1$, and a number of universal constant), which are hard to interpret. I think the paper would be improved if there is a longer/higher-level paragraph of discussion on the analysis of Theorem 4.2 to help the reader to parse it. At the moment there is just a brief discussion on lines 267-268.

Minor comments:
- I found it hard to spot the definitions, I would suggest bolding them so it's slightly easier to read (up to your preference)

---

> ### Author Response · Authors · 2022-08-01
> **Regarding Assumption 4.1, Theorem 4.2, ratio-cut plots, and the fairness tradeoff**
>
> **Update 1:** All the changes mentioned below have now been added to the revised manuscript. Please look for the changes highlighted in brick-red color.
>
> Thank you for your comments and constructive feedback. We hope that our responses below address your concerns.
>
> 1. **Regarding Assumption 4.1 [answer shared with Reviewer 7uut]:**
> As Reviewer 7uut has correctly pointed out, proving such results is a very challenging problem that often requires strong assumptions. For example, Kleindessner et al. 2019 make equally strong assumptions even though we significantly generalize their results.
> As mentioned in L231, Assumption 4.1 is sufficient (but not necessary) to guarantee the existence of representation aware ground truth clusters. It also allows us to compute eigenvalues and eigenvectors of the laplacian matrix in the expected case, which is a crucial step in the analysis of spectral algorithms [Rohe et al. 2011; Lei and Rinaldo 2015]. Relaxing Assumption 4.1 requires finding alternative solutions to these problems and we do believe that it is an interesting direction for future work (L341-342).
> Finally, we wish to re-emphasize that Assumption 4.1 is not necessary in practice (please see Remark 5 and Section 5).
>
> 2. **Regarding Theorem 4.2 and interpretation of $\lambda_1$ [answer shared with Reviewer N1aR]:**
> We begin with $\lambda_1$. $\lambda_1$ is the largest eigenvalue of a slightly modified expected adjacency matrix under the $\mathcal{R}$-PP model (the largest eigenvalue of the expected adjacency matrix is given by $\lambda_1 - p$ (see eq. (15) and Lemma D.2)).
> All three assumptions in Theorem 4.2 essentially control the minimum rate at which $\lambda_1$ must grow with $N$. For instance, the first two assumptions can be replaced by a simpler but slightly stronger (yet practical) condition: $\lambda_1 - p = \omega(\sqrt{p N \ln N})$. This, for example, is satisfied when $K = O(\ln N)$ and $d = O(\ln N)$ which is not unrealistic.
> The third assumption is more involved because of the presence of the eigengap $\gamma$. While this assumption too controls the rate of growth of $\lambda_1$, it does so in context of the community structure contained in the $\mathcal{Q}^{-1} \mathbf{Y}^T \mathcal{L} \mathbf{Y} \mathcal{Q}^{-1}$, as encoded by the eigengap $\gamma$ (please see Theorem 4.2 for the definition of these symbols). So while $\lambda_1$  must increase, it must not be at the expense of the community structure (for instance, when p=q=r=s=1).
> We will use the extra space in the camera ready version to add this discussion to the paper.
>
> 3. **Regarding plots with respect to ratio-cut:**
> Please see our common response titled “Additional plots”.
>
> 4. **Can both ratio-cut and representation-awareness be maximized simultaneously?**
> Note that ratio-cut is only dependent on the structure of the similarity graph $\mathcal{G}$ whereas balance is computed using the structure of the representation graph $\mathcal{R}$. Therefore, in general, it is not possible to simultaneously maximize both of these quantities. In fact, as we describe in L215-218, ratio-cut and balance are at odds with each other for graphs sampled from $\mathcal{R}$-PP, thereby ensuring that $\mathcal{R}$-PP presents a hard problem instance to our algorithm.
>
> We will also make the definition headers bold so that it is easier to spot them. Thanks for pointing this out.

---

> > ### Author Response · Authors · 2022-08-08
> > **Regarding our replies to questions raised by Reviewer 1o3a**
> >
> > We have given detailed responses to several important questions raised by Reviewer 1o3a. We request the reviewer to let us know if they have any further questions. We will be happy to provide answers. This will really help us to improve the paper. Thanks again.

---

### Official Review · Reviewer_7uut · 2022-07-11

**Rating:** 6
**Confidence:** 3
**Soundness:** 3 good
**Presentation:** 4 excellent
**Contribution:** 4 excellent

**Summary:**

The paper studies spectral algorithms for clustering under individual fairness constraints. This is the first spectral algorithm for this setting; the only previous fair spectral clustering algorithm was by Kleindessner et al. (ICML'19) and considered population level fairness.

The individual fairness constraint studied in the submission is newly proposed. The new problem assumes that besides the standard similarity graph (in which the clusters shall minimize the ratio-cut), there is also a *representation graph*. An edge in the representation graph indicates the similarity of two users. Now a novel constraint is proposed, which requires that *for each vertex* its neighbors (in the representation graph) are assigned to the clusters "evenly" among the clusters. The goal is to minimize the ratio-cut of the clusters in the similarity graph. It is shown that the previous results of Kleindessner et al. (ICML'19) and Chierichetti et al. (NeurIPS'17) are special cases of this setting.

Then spectral algorithms for this problem are derived. It is shown that under a newly introduced random graph model, which is akin to the SBM, these algorithms classify almost all vertices correctly.

**Questions:**

The theorems require a (1+eps)-approximation algorithm for k-Means. However, k-Means is APX-hard. The cited algorithm from reference [28] only works when k is constant. I think this should be discussed in more detail.

**Limitations:**

I think the limitations and the potential impact are sufficiently discussed.

**Strengths And Weaknesses:**

The paper is very well-written and easy to follow. The proofs for the theoretical are non-trivial. The newly proposed model for individual fairness is quite intriguing since it generalizes existing notions but at the same it "requires similar individuals to be spread across different clusters" (see Lines 156 and 157). The latter is somewhat counterintuitive compared to more classic clustering objectives. The reason is that in the proposed problem (Equation 7 in the paper), the objective is to minimize the ratio-cut in the similarity graph (similar to typical clustering tasks) but the constraint forces to split up the neighborhoods from the representation graph (unlike typical clustering tasks). I find this quite interesting and could imagine that this setting finds more applications in the future.

The paper also has some weaknesses. The main weakness is certainly that the theoretical results only hold under a relatively strong assumption (Assumption 4.1 in the paper), which stipulates that every vertex has degree d and it must have exactly d/K neighbors from each cluster in the final clustering. This is certainly a bit unrealistic, but deriving theoretical results for these types of problems is quite difficult, so I do not hold it against the paper too much.

In conclusion, I value the merits of the paper higher than its weaknesses. I believe the newly proposed problem raises enough interesting questions to merit further study.

---

> ### Author Response · Authors · 2022-08-01
> **Assumption 4.1 and the complexity of k-Means**
>
> **Update 1:** All the changes mentioned below have now been added to the revised manuscript. Please look for the changes highlighted in brick-red color.
>
> Thank you for carefully reading our paper and appreciating our work. Below, we address your concerns:
>
> 1. **Regarding Assumption 4.1 [answer shared with Reviewer 1o3a]:**
> As you have correctly pointed out, proving such results is indeed a very challenging problem that often requires strong assumptions. For example, Kleindessner et al. 2019 make equally strong assumptions even though we significantly generalize their results.
> As mentioned in L231, Assumption 4.1 is sufficient (but not necessary) to guarantee the existence of representation aware ground truth clusters. It also allows us to compute eigenvalues and eigenvectors of the laplacian matrix in the expected case, which is a crucial step in the analysis of spectral algorithms [Rohe et al. 2011; Lei and Rinaldo 2015]. Relaxing Assumption 4.1 requires finding alternative solutions to these problems. We do believe that this is an interesting direction for future work (L341-342).
> Finally, we wish to re-emphasize that Assumption 4.1 is not necessary in practice (please see Remark 5 and Section 5).
>
> 2. **Regarding k-Means:**
> Again, you are absolutely right. The algorithm in Kumar et al. 2004 gives a linear time approximation only when $K$ is constant. For non-constant values of $K$, we still get a $(1+ \epsilon)$-approximation, though not in linear time. This issue is faced by several variants of spectral clustering when $K$ increases with $N$ (e.g., see L308-309).
> Several practical implementations of $k$-Means have a linear average time complexity in both $N$ and $K$ (e.g., a variant in scikit-learn), though they no longer guarantee a $(1+\epsilon)$-approximation. These implementations can be used in practice whenever necessary. We will add this discussion to the paper. Thanks for pointing it out.

---

> > ### Comment · Reviewer_7uut · 2022-08-04
> > **Thank You!**
> >
> > Thank you for your answer and the revised version of the paper.

---

### Author Response · Authors · 2022-08-01
**Additional Plots**

**Update 1:** All the changes mentioned below have now been added to the revised manuscript. Please look for the changes highlighted in brick-red color.

We thank the reviewers for an overall positive evaluation of the paper. This comment pertains to questions regarding our experiments.

We present three types of plots here: **(i)** Separate plots for ratio-cut and average balance (Reviewers 103a and yXMv), **(ii)** Empirical validation of time complexity (Reviewer yXMv), and **(iii)** Another experiment on a real-world network (Reviewer yXMv). All of this information will be added to the supplementary material in the next revision.

Plots are hosted on imgur by an anonymised account.

&nbsp;

### Separate plots for ratio-cut and average balance:
We decided to show the number of mistakes (for d-regular $\mathcal{R}$) and the ratio of average balance to ratio-cut (for SBM based $\mathcal{R}$ and real-world network) on y-axis because these quantities adequately convey our main point: our algorithm finds high-quality fair clusters.

However, as asked by the reviewers, we have separated the ratio-cut and average balance for all the plots in the paper. Here is just a small subset for the sake of clarity (normalized variant used in all cases). If the reviewers want, we will be happy to post all of our new plots. In any case, all the plots will be added to the supplementary material in the revision.
1. **For $d$-regular graphs:** Average balance vs $N$ [[https://imgur.com/DIH9m4V](https://imgur.com/DIH9m4V)], Ratio-cut vs $N$ [[https://imgur.com/QfskxrI](https://imgur.com/QfskxrI)]
2. **For SBM-based $\mathcal{R}$:** Average balance [[https://imgur.com/WmylN0d](https://imgur.com/WmylN0d)], Ratio-cut [[https://imgur.com/LugTJqv](https://imgur.com/LugTJqv)]
3. **For FAO Trade Network:** Average balance [[https://imgur.com/kiHmtIT](https://imgur.com/kiHmtIT)], Ratio-cut [[https://imgur.com/reZXqfz](https://imgur.com/reZXqfz)]

As expected, our algorithm is competitive with (or better than) NFairSC (Kleindesser et al. 2019) in terms of Ratio-Cut (lower is better) while providing higher individual balance.

&nbsp;

### Time complexity validation
The plot here [[https://imgur.com/uhwoSsf](https://imgur.com/uhwoSsf)] presents the time in seconds (on y-axis) required by our algorithms as a function of the number of nodes in the graph. We used SBM based representation graphs for these plots ($K=4$, $R = 0.5N$) but the results are similar for other configurations. The time shown includes getting a low-rank approximation for $\mathcal{R}$. This corroborates our $O(N^3)$ time complexity analysis from Remark 2.

&nbsp;

### Another real-world network
We want to stress that our primary contributions are theoretical in nature. In the limited rebuttal period, we were able to run experiments with one more real-world network - Air Transportation Multiplex Network (Cardillo et al. 2013). Having said that, we will continue looking for other real-world datasets that can be used in our experiments.

In this network, nodes correspond to airports, edges correspond to direct connections, and layers correspond to airlines. We took three designated “major” airlines and constructed a similarity graph by taking the union of the edges in these layers. Similarly, we constructed the representation graph by considering three “lowcost” airlines. Nodes that were isolated in either of the similarity/representation graphs were dropped. The resulting graphs have 106 nodes.

The plots linked show ratio-cut [[https://imgur.com/Xpkr9TH](https://imgur.com/Xpkr9TH)] and average balance [[https://imgur.com/H3JNuGX](https://imgur.com/H3JNuGX)] for a particular choice of $K$ (again, normalized variant). Similar results hold for other choices of $K$ as well. These plots have the same semantics as the FAO Trade Network plots above. As before, we see higher balance at a competitive ratio-cut.

---

### Meta-Review · Area_Chair_BF1V · 2022-08-24

**Recommendation:** Accept
**Confidence:** Certain

**Metareview:**

The paper presents a new interesting spectral generalization of fair clustering. The new notion captures many of the previously introduced notion and unify few of them. The authors present also an algorithm for the new notion.

Overall the paper presents interesting idea and it is nicely written so it would be a nice contribution to NeurIPS program.

The suggestion of the AC is to accept the paper as a poster.

**Award:**

No

---

### Decision · Program_Chairs · 2022-09-14

Accept